REGISTERED REPORT PROTOCOL

# Policies for healthy ageing in response to climate change: Protocol of a systematic review

Nestor Asiamah[1,2,3,4]*, Bernard Opoku Ofosu[5], Yaw Jnr Effah-Baafi[6], Kofi Awuviry-Newton[7], Jacob Nkansah[8], Bernadette Saunders[9]

1 Division of Interdisciplinary Research and Practice, School of Health and Social Care, Colchester, Essex, United Kingdom, 2 International Public Health Management Programme, University of Europe for Applied Sciences, Iserlohn, Germany, 3 Research Faculty, Berlin School of Business and Innovation, Berlin, Germany, 4 Africa Center for Epidemiology, Department of Geriatrics and Gerontology, Accra, Ghana, 5 School of Public Health, University of Ghana, Legon-Accra, Ghana, 6 Kumasi, Centre for Collaborative Research in Tropical Medicine, Kwame Nkrumah University of Science and Technology, Kumasi, Ghana, 7 Department of Allied Health, College of Sport, Health and Engineering, Victoria University, Melbourne, Victoria, Australia, 8 Department of Education Policy and Leadership, The Education University of Hong Kong, Ting Kok, Hong Kong, 9 Department of Allied Health, College of Sport, Health and Engineering, Victoria University, Melbourne, Victoria, Australia

* n.asiamah@essex.ac.uk

## Abstract

Climate change is a global phenomenon affecting every segment of the population. Yet, older adults are more vulnerable to climate change events (e.g., floods, heatwaves, landslides) owing to their functional limitations. Understandably, stakeholders have called for healthy ageing policies that enable older adults and individuals in the general population to maintain wellbeing despite climate change. This review aims to describe healthy ageing policies adopted or recommended in response to climate change. Eight databases (i.e., CINAHL, Cochrane library, ProQuest, PsycINFO, Google Scholar, Web of Science, Scopus, and MEDLINE) will be searched to identify relevant studies. Materials published anywhere in English to date will be included in the review. The Critical Appraisal Skills Programme (CASP) or Joanna Briggs Institute (JBI) checklist will be employed to assess the quality of studies. A narrative synthesis will be adopted to present the results. This review will highlight groups targeted with healthy ageing policies and describe policies in use or recommended. It will proffer implications for practice, research, and sustainability.

## Introduction

Climate change accompanies extreme weather conditions, (e.g., flooding, hurricanes, and heat waves) that adversely impact communities. It threatens biodiversity, increases the risk of mortality, and makes communities more vulnerable to respiratory diseases [1,2]. Although it affects countries and regions in different ways, researchers

**Data availability statement:** All relevant data are within the manuscript and its Supporting Information files.

**Funding:** The author(s) received no specific funding for this work.

**Competing interests:** The authors have declared that no competing interests exist.

[3,4] agree that climate change is globally felt, affects some segments of the population more strongly, and needs a concerted rapid response from stakeholders.

One of the segments of the population most affected by climate change events such as heatwaves and flooding is older adults [5,6]. Owing to their physiological limitations, older adults are less likely to cope with disasters caused by climate change, and this group loses essential services and support following the above events [7–9]. Older adults with comorbidities (e.g., asthma, diabetes, and hypertension) and functional limitations are less likely to survive extreme weather events. In their recognition of how enormous the impact of climate change on individuals is, researchers [1,3,10] have called for interventions aimed at enabling older adults to cope with the worsening climate crisis. This call is timely owing to rapid population ageing.

Population ageing refers to a global demographic shift characterised by people living longer due to declining fertility rates and increasing life expectancy [11]. Because of population ageing, the population of older adults aged 65 years or older is expected to more than double by 2050 [11,12]. Although population ageing is more rapid in high-income countries [13], it is being experienced by every country. Thus, population ageing and climate change both prevail on a global scale, which means their respective impacts can reach every jurisdiction.

Like climate change, population ageing affects everyone. Suffice it to say that people of all ages are subject to the ageing process. As people age, they lose the ability to perform physical and social activities and are, therefore, less likely to manoeuvre daily challenges accompanied by climate change. If so, climate change preparedness programmes should include interventions enabling all ageing people to cope with or navigate climate change events. Research in the past decades shows increased advocacy for healthy ageing policies in response to climate change [14,15]. While some studies report policies already developed for fostering healthy ageing in contexts experiencing climate change events, others have reported policies recommended by experts for the same purpose [14,16].

Healthy ageing is a process in which functional capacity is maintained to enable well-being in old age [17]. As such, healthy ageing policies against climate change would enable individuals to maintain functional ability and health over the life course. Functional ability encompasses autonomy, resilience, and successful coping with crises [18]. The above definition suggests that the result of healthy ageing is well-being in later life. Hence, the ideal policies for healthy ageing in response to climate change should be targeted at every ageing individual.

What is unclear in the literature is whether policies for healthy ageing in response to climate change are targeted at the general population or specific groups (e.g., older adults and people with disability). Ascertaining who is targeted by stakeholders with the foregoing policies would unfold implications for improved action. Another issue is that policies already developed through formal legislation (category 1) and those recommended by experts for implementation (category 2) are obscured in the literature. A clear identification and analysis of these two categories of policies would facilitate their adoption and encourage research into their suitability.

## Objectives

The objectives of this proposed systematic review are (1) to describe policies for healthy ageing adopted (category 1) or recommended in the literature (category 2) in response to climate change, and (2) to describe populations or groups targeted with these policies.

## Methods

### Review protocol and registration

This protocol is registered (number CRD42025641151) with PROSPERO [19], which is an international prospective register of systematic reviews and meta-analyses. It adheres to the protocol version of the Preferred Reporting Items for Systematic Reviews and Mata-Analyses (PRISMA-P) [20]. Appendix 1 shows a sample PRISMA-P. Subsequently, the systematic review will follow the Reporting Items for Systematic Reviews and Mata-Analyses (PRISMA) flow-chart [21,22]. Any protocol amendments carried out later will be reported in the systematic review.

### Types of studies

Peer-reviewed papers and grey literature published in English to date will be included in this review. Although systematic reviews may focus on peer-reviewed articles, we plan to include grey literature (e.g., policy analysis and white papers) because this is more likely to contain information about existing or recommended policies. All types of designs (e.g., experimental, observational, and quasi-experimental designs) and publications (e.g., commentary, policy analysis, narratives) will be included.

### Population and participants

Only studies focused on ageing and discussing healthy ageing policies will be included. The studies or documents will not be restricted to any period or geographical region. Since ageing is a lifelong process that affects everyone, studies on the general population or specific groups that report policies on healthy ageing will be included. Studies or documents will not be included if they do not report policies in response to climate change or global warming.

### Types of interventions

Healthy ageing policies can be recommended, described, or reported in any type of study, intervention, or publication; hence, studies included in this review should employ any research design. Even so, we expect Delphi studies and research utilising data from expert interviews and focus group discussions to be more likely sources of existing or recommended policies.

### Types of comparators

Comparators will be existing policies that act as a barrier to the adoption of healthy ageing policies. These comparators are policies discouraging stakeholders to adopt or roll out healthy ageing policies.

### Types of outcome measures

The primary outcome of this review is describing the policies that are being rolled out or have been recommended against climate change and its events to support people to maintain health as they age.

We will identify or assess these policies based on whether they are already being rolled out or are merely recommended. "Policies being rolled out" are a framework of guidelines backed by law (i.e., legislation) and are currently being implemented by a community, country, or region. "Policies recommended" are a framework of guidelines proposed by

experts but are not legally adopted in the given context. Outcome measures will include a description of groups targeted with the policies (e.g., whether policies are targeted at only older adults or the general population).

## Search strategy

### Electronic search

Electronic searches will be performed in eight standard databases: Web of Science, SCOPUS, MEDLINE, PsycINFO (accessed via EBSCO Host), Cochrane Library, ProQuest, Google Scholar, and CINAHL Ultimate (accessed via EBSCO Host). Searches will be based on (1) healthy ageing, (2) climate change, and (3) policies. Terms relating to these key-words will be identified by exploring Medical Subject Headings (MeSH) terms at MEDLINE (via PubMed). Identification of MeSH terms will be done by two independent individuals in the review team and any disagreement will be resolved by them through a discussion. A sample of the search strategy generated is shown in Appendix 2. The sensitivity of the search strategy will be examined by assessing whether it retrieves five key articles [23] that were previously identified with Elicit, a digital tool powered by artificial intelligence [24].

### Non-electronic search

Some relevant articles or documents may not be available online. To identify and include such documents, we will search the printed literature catalogues of the University of Essex Library. This library has information about all its printed documents (e.g., articles, book chapters, and grey literature) on a digital platform where researchers can search for printed literature. We will perform an electronic search for relevant printed documents on this platform with our search strategy and subsequently request relevant documents found from the library.

### Document selection

All documents found in the eight electronic databases will be exported to EndNote (Thompson Reuters) where duplicate documents will be removed with the EndNote auto-duplication function. Because this function partially removes duplicates [25], we will remove any other duplicates with the hand-searching technique. In this vein, references will be ordered by authors' first names and titles, allowing for the identification of duplicates skipped earlier.

### First stage screening

Two independent reviewers will initially screen all titles and abstracts after removing duplicates. This process is aimed at separating studies that are potentially eligible from those not eligible. To identify "potentially eligible" studies, the following procedures will be followed:

**Study participants.** For materials to be considered potentially eligible, their titles or abstracts should clearly show that the study participants were individuals of the general population (often people aged 18 years or older). In the literature, 50 years is used as the minimum old age [26], so studies including older adults aged 50 years or older will be potentially eligible. Such materials will be seen as studies focused on older adults.

**Intervention.** All interventions (e.g., RCT and longitudinal studies), reviews, commentaries, or publicly available material that report relevant healthy ageing policies will be deemed potentially eligible. Titles or abstracts must clarify whether any policy recommendations, acknowledgements, or discussions are in the study.

**Study characteristics.** Potentially eligible studies must be published in English to date. Such studies can be peer-reviewed materials or grey literature (e.g., policy analysis, white papers, and policy briefs).

After documents are screened as discussed above, both independent reviewers will meet to identify and resolve any disagreements. If a consensus cannot be reached by them, a third reviewer will be invited to help reach a consensus. The inter-rater reliability between the two reviewers will be calculated with Cohen's kappa statistic to establish a final consensus [27].

## Second stage screening

At this stage, full texts of all potentially eligible documents will be reviewed. Subsequently, 10% of documents deemed eligible will be checked by a second reviewer. Following the PRISMA flowchart [20,22], the various stages of the screening and selection process will be illustrated and reasons for removing some documents will be stated.

## Data extraction and management

Two reviewers will independently extract data from included documents and enter the data into a piloted extraction sheet (please see Appendix 3). Data in the two filled sheets will subsequently be checked by a third reviewer to ensure consistency. Any differences in the two sheets will be discussed and resolved by all reviewers. If the difference is unresolved, the authors of the papers will be contacted, enabling the reviewers to reach an agreement or consensus [23]. If the required data are missing in the article or are not clarified, the authors of the papers will be contacted.

## Data items

The following information will be extracted from each article based on our review objectives: author, publication year, journal name, study design, study setting, type of participants, sample size, country of study, sample demographics (e.g., gender, age, and ethnicity), healthy ageing policies recommended against climate change, healthy ageing policies already adopted against climate change, and policies that discourage the adoption of healthy ageing policies (i.e., counter policies).

## Risk of bias assessment

The risk of bias and quality of all studies will be assessed with the CASP (Critical Appraisal Skills Programme) or JBI (Joanna Briggs Institute) checklist, depending on which of them is most suited for the studies included [28]. Two reviewers in the team will independently conduct a risk assessment using CASP or JBI, and any disagreements will be resolved through a meeting between the independent reviewers.

## Data synthesis

To identify all healthy ageing policies against climate change, the search strategy utilised in this systematic review is made robust and open. Data synthesis will be based on the review objectives and readers' potential information needs. Extracted data will, thus, be organised into four categories: (1) bibliographic information of the studies, which will enable readers to identify key attributes of the review; (2) healthy ageing policies adopted; (3) healthy ageing policies recommended, (4) groups targeted by the policies, and (5) counter policies. A narrative synthesis [23] will be utilised to organise the studies since the review aims to identify and interpret available policies in their contexts. As part of the synthesis, a table will be created to map all identified policies onto their target groups, contexts (countries), and stage of implementation of adopted policies (i.e., whether the policy is already being implemented or not). Figures will also be used to summarise bibliographic information.

After organising the studies as discussed above, the final document reporting the outcomes of the review will be prepared by the reviewers following the PRISMA guideline [22]. Experts in the field of healthy ageing, preferably corresponding authors of the papers included, will be invited to review our mapping (table) and confirm if the information in the table reflects the contexts they studied or reported in their papers. In situations where the authors are not available, other

experts in those contexts will be invited. All experts and corresponding authors invited will be given access to documents included in the review.

## Discussion and conclusion

Older adults are disproportionately affected by climate change, so programmes aimed at meeting their unique needs in contexts experiencing extreme climate events are essential. Researchers [29–31] have acknowledged a need for countries to adopt healthy ageing policies that protect older adults from climate crises, and the literature shows a range of healthy ageing policy responses to climate change [32,33]. This systematic review compiles healthy ageing policies that are cognisant of climate change, giving stakeholders insight into the purpose of these policies and who they are targeted at. A systematic review allows us to demonstrate the highest level of rigour in our search, scrutiny, and reporting of the relevant evidence.

This review will reveal contexts where the development of healthy ageing policies in response to climate change should be prioritised. By compiling evidence on a global scale, this review will depict the geographic scope of ageing policy responses to the climate crisis. The review may provide an understanding of the role of healthy ageing policies in public health promotion during climate change. Finally, this review will highlight areas where future research and policy dialogues are needed for fostering healthy ageing.

## Supporting information

**S1 Appendix 1. This is the S1 Appendix 1 Title.**
(DOCX)

**S2 Appendix 2. This is the S2 Appendix 2 Title.**
(DOCX)

**S2 Appendix 3. This is the S3 Appendix 3 Title.**
(DOCX)

## Author contributions

**Conceptualization:** Nestor Asiamah, Bernard Opoku Ofosu, Yaw Jnr Effah-Baafi, Kofi Awuviry-Newton.

**Data curation:** Bernard Opoku Ofosu.

**Methodology:** Nestor Asiamah, Yaw Jnr Effah-Baafi.

**Project administration:** Nestor Asiamah, Kofi Awuviry-Newton.

**Resources:** Nestor Asiamah.

**Software:** Nestor Asiamah.

**Supervision:** Nestor Asiamah.

**Validation:** Nestor Asiamah, Bernard Opoku Ofosu, Yaw Jnr Effah-Baafi, Kofi Awuviry-Newton, Jacob Nkansah, Bernadette Saunders.

**Visualization:** Nestor Asiamah, Bernard Opoku Ofosu, Yaw Jnr Effah-Baafi, Kofi Awuviry-Newton, Jacob Nkansah, Bernadette Saunders.

**Writing – original draft:** Nestor Asiamah.

**Writing – review & editing:** Nestor Asiamah, Bernard Opoku Ofosu, Yaw Jnr Effah-Baafi, Kofi Awuviry-Newton, Jacob Nkansah, Bernadette Saunders.

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
