## [Decision Letter · Decision Letter 0]

11 Mar 2025

PONE-D-25-06987Policies for Healthy Ageing in Response to Climate Change: Protocol of a Systematic ReviewPLOS ONE

Dear Dr. Asiamah,

Thank you for submitting your manuscript to PLOS ONE. After careful consideration, we feel that it has merit but does not fully meet PLOS ONE’s publication criteria as it currently stands. Therefore, we invite you to submit a revised version of the manuscript that addresses the points raised during the review process.

 Please review and address the comments provided by both reviewers available at the end of this email.. 

We look forward to receiving your revised manuscript.

Kind regards,

LS Katrina Li

Academic Editor

PLOS ONE

Journal Requirements:

2. In your cover letter, please confirm that the research you have described in your manuscript, including participant recruitment, data collection, modification, or processing, has not started and will not start until after your paper has been accepted to the journal (assuming data need to be collected or participants recruited specifically for your study). In order to proceed with your submission, you must provide confirmation.

3. As required by our policy on Data Availability, please ensure your manuscript or supplementary information includes the following:

4. We note that you have indicated that there are restrictions to data sharing for this study. PLOS only allows data to be available upon request if there are legal or ethical restrictions on sharing data publicly. For more information on unacceptable data access restrictions, please see http://journals.plos.org/plosone/s/data-availability#loc-unacceptable-data-access-restrictions . 

Reviewers' comments:

Reviewer's Responses to Questions

**Comments to the Author**

1. Does the manuscript provide a valid rationale for the proposed study, with clearly identified and justified research questions?

Reviewer #1: Yes

Reviewer #2: Partly

2. Is the protocol technically sound and planned in a manner that will lead to a meaningful outcome and allow testing the stated hypotheses?

Reviewer #1: Yes

Reviewer #2: Partly

3. Is the methodology feasible and described in sufficient detail to allow the work to be replicable?

Reviewer #1: Yes

Reviewer #2: Yes

4. Have the authors described where all data underlying the findings will be made available when the study is complete?

Reviewer #1: Yes

Reviewer #2: Yes

5. Is the manuscript presented in an intelligible fashion and written in standard English?

Reviewer #1: Yes

Reviewer #2: Yes

6. Review Comments to the Author

You may also provide optional suggestions and comments to authors that they might find helpful in planning their study.

Reviewer #1: Well done on your Protocol of a Systematic Review. Please find below specific comments to address.

INTRODUCTION

1. Reference needed after first sentence

2. Reference needed second paragraph, third sentence

3. Insert "globally" to third paragraph sentence- Because of population ageing, the population of older adults aged 65 years or older "globablly" is expected to more than double by 2050.

4. Reference needed third paragraph, third sentence

5. Remove paragraph 4 related to people of all ages as unclear why this is needed?

6. Remove from healthy ageing paragraph sentence "Hence, the ideal policies for healthy ageing in response to climate change should be targeted at every ageing individual"

7. Include at end of healthy ageing paragraph "Research in the past decades shows increased advocacy for healthy ageing policies in response to climate change [14, 15]. While some studies report policies already developed for fostering healthy ageing in contexts experiencing climate change events, others have reported policies recommended by experts for the same purpose [14, 16]."

METHODS

8. Correct spelling "Meta-Analyses" in (Review protocol and registration)

9. Include "also" in (Types of studies)- Although systematic reviews may focus on peer-reviewed articles, we plan to "also" include grey literature

10. Provide exclusion criteria for (Types of studies)

11. Remove "global warming" as this is not mentioned in introduction

12. Unclear if the current search strategy will identify (Types of comparators)- barriers to adoption healthy ageing, please consider describing how this will be achieved

13. Include "people with a disability" (Types of outcome measures)- brackets in final sentence

14. (First stage screening)- Study participants= Remove "often people aged 18 years or older"

15. Include detail information about literature uses 50 years as the minimum old age in Introduction, as currently you talk about older adults as 65 years or older

16. Study Participants= Remove "will be potentially eligible. Such materials"

17. (Risk of Bias Assessment)= Ensure you write in full before using abbreviations e.g. Critical Appraisal Skills Programme (CASP) or Joanna Briggs Institute (JBI)

18. Include "either" to first sentence Risk of Bias Assessment- The risk of bias and quality of studies will be assessed with "either" the

DISCUSSION AND CONCLUSION

19. Keep consistent terminology throughout paper, change programmes in first sentence to "policies"

20. Unclear who the specific stakeholders are? (e.g. general population/individuals, parliament, organisations, academics etc)

21. Reword final sentence suggestion "Finally, this review will highlight areas where future research and policy dialogues are needed to foster successful implementation of healthy ageing policies."

Reviewer #2: 1. The introduction provides a partly valid rationale for the proposed study, clearly justifying research questions; however, some parts need more clarity:

- Some broad statements should be backed by specific examples/evidence. For example, "Research in the past decades shows increased advocacy for healthy ageing policies in response to climate change [14, 15]" should be more specific such as what kind of policies or frameworks have been developed and in which contexts (e.g., WHO frameworks, national-level policies).

- The phrase “climate change accompanies” is inappropriate. Climate change increases the frequency and intensity of extreme weather events rather than simply accompanying them.

- The pathways through which climate change affects health are not well defined. "It threatens biodiversity, increases the risk of mortality, and makes communities more vulnerable to respiratory diseases." The phrase “makes communities more vulnerable” is too broad and lacks specific mechanisms. Data should support the statement.

2. The search strategy is well-structured, but some important keywords are missing in the search strings.

In your appendix 2 "Search strings and strategy generated across key search databases" consider including the following terms:

- S1 (population): Add: older adults, elderly, aging population, successful aging, aged individuals, senior citizens, older people, late-life adults

- S2 (climate): environmental change, extreme weathers (may be specific to storms, floods, heatwaves, heat stress, rising temperatures, extreme heat, cyclones, disasters, cold waves, extreme cold, winter storms, urban heat islands, rising sea levels)

- S3 (policy): climate change adaptation, governance, intervention, strategy, climate mitigation, emergency preparedness, disaster risk reduction

7. PLOS authors have the option to publish the peer review history of their article (what does this mean? ). If published, this will include your full peer review and any attached files.

**Do you want your identity to be public for this peer review?** For information about this choice, including consent withdrawal, please see our Privacy Policy .

Reviewer #1: No

Reviewer #2: **Yes: ** Nu Quy Linh Tran

---

## [Author Response · Author response to Decision Letter 1]

21 Mar 2025

The response to reviewers is attached as a file

---

## [Editor Report · Decision Letter 1]

2 Apr 2025

Policies for Healthy Ageing in Response to Climate Change: Protocol of a Systematic Review

PONE-D-25-06987R1

Dear Dr. Asiamah,

Thank you for submitting your revised manuscript and addressing comments/ concerns raised by the reviewers. We’re pleased to inform you that your manuscript has been judged scientifically suitable for publication and will be formally accepted for publication once it meets all outstanding technical requirements.

Kind regards,

LS Katrina Li

Academic Editor

PLOS ONE
---

## [Editor Report · Acceptance letter]

PONE-D-25-06987R1

PLOS ONE

Dear Dr. Asiamah,

I'm pleased to inform you that your manuscript has been deemed suitable for publication in PLOS ONE. Congratulations! Your manuscript is now being handed over to our production team.

Kind regards,

on behalf of

Dr. LS Katrina Li

Academic Editor

PLOS ONE